# Towards an Inclusive Approach to Forest Management: Highlight of the Perception and Participation of Local Communities in the Management of *miombo* Woodlands around Lubumbashi (Haut-Katanga, D.R. Congo)

Dieu-donné N'tambwe Nghonda [1,2,*], Héritier Khoji Muteya [1,2], Bill Kasongo Wa Ngoy Kashiki [1], Kouagou Raoul Sambiéni [3,4], François Malaisse [2], Yannick Useni Sikuzani [1], Wilfried Masengo Kalenga [1] and Jan Bogaert [2]

1 Unité Écologie, Restauration Écologique et Paysage, Faculté des Sciences Agronomiques, Université de Lubumbashi, Lubumbashi P.O. Box 1825, Democratic Republic of the Congo
2 Axe Biodiversité et Paysage, Université de Liège—Gembloux Agro-BioTech, 5030 Gembloux, Belgium
3 Ecole Régionale Postuniversitaire d'Aménagement et de Gestion Intégrée des Forêts et Territoires Tropicaux (ERAIFT), Kinshasa P.O. Box 15373, Democratic Republic of the Congo
4 Faculté d'Architecture, Université de Lubumbashi, Lubumbashi P.O. Box 1825, Democratic Republic of the Congo
* Correspondence: nghondan@unilu.ac.cd; Tel.: +243-97-25-700-26

**Abstract:** The misappropriation of sustainable forest programs by local communities and the under-utilization of their knowledge are major impediments to the mitigation of deforestation. Within this context, participation has become a principle used in almost all interventions. It is important to evaluate the practices in this area to ensure better involvement of local communities. This survey examined the perception and participation of local communities in the management of *miombo* woodlands, based on semi-structured questionnaire surveys involving 945 households in 5 villages in the Lubumbashi rural area. The results reveal that local communities perceive soil fertility loss and deforestation as major environmental challenges in their area. This perception remains largely influenced by their socio-demographic factors such as respondents' age, seniority in the villages, and level of education. To mitigate deforestation, the rare actions of provincial public services and non-governmental organizations are focused on the sustainable exploitation of *miombo* woodlands through the development of simple management plans, reforestation, and forest control. These activities are sparse and poorly inclusive of scientific findings and the priorities of local communities. These justify poor community participation, particularly in the actions of provincial public services. For a better appropriation of sustainable forest management plans and to reinforce *miombo* woodlands' resilience to anthropogenic pressures, based on these findings, we recommend a concerted and inclusive approach to forest planning.

**Keywords:** perception; forest policy; *miombo* woodland; deforestation; public administration; development agency; participatory management

## 1. Introduction

Forests are among the most biodiverse ecosystems on the planet [1]. As such, they provide many ecosystem services, including climate regulation, provision of timber, fuelwood, and various non-timber forest products [2]. With the global human population increasing from 1.86 billion in 1920 to eight billion in 2022, the demand for various forest products is mounting anthropogenic pressure on forest resources, leading to deforestation [3]. Deforestation has been counted among the critical environmental issues in recent decades, attracting the attention of both scientists and policymakers [4].

Indeed, forests that occupied four billion hectares [3], or nearly 30% of the Earth's land surface, recorded a loss of 178 million hectares between 2015 and 2020 [5]. The highest forest cover losses are observed in Southeast Asia (−4.5 million ha per year) and Central Africa (−6.9 million ha per year) [6]. Regarding Central Africa, out of a forest area of nearly 300 million hectares [7], an annual loss of 1.79 million hectares was noted between 2015 and 2020 in the Congo Basin [8], mainly due to inadequate forest governance [9]. This form of governance is embodied by governments' persistent centralization of responsibilities, and inadequate and poorly enforced laws [10]. This results in the unsustainable exploitation of forest resources, deforestation, and forest degradation [9].

The need for improved forest governance remains urgent in countries such as the Democratic Republic of Congo (DRC), given its highest annual deforestation rate in the Congo Basin, 0.26% between 2005 and 2010 [11], in conjunction with weak environmental law enforcement [12]. The southeastern region of DRC, where the *miombo* woodlands remain the most represented forest type [13], does not escape this reality. *Miombo* woodlands are a type of open forest characterized by the predominance of plant species belonging to the genera *Brachystegia*, *Julbernardia*, and *Isoberlinia* [14]. They covered more than 70% of the Katangese territory in 2000, declined to nearly 43% in 2010 [15], and are predicted to disappear in the vicinities of large human settlements by 2090 [16]. Yet, *miombo* woodlands, which support the survival of more than two million people in Haut-Katanga [13], are considered an important carbon reservoir and prioritized for conservation due to high floristic diversity and endemism [17].

With a population that doubles every 15 years, Lubumbashi, the capital city of Haut-Katanga Province, currently has nearly three million inhabitants [18]. This amplifies fuelwood demands [19] due to limited household access to hydroelectricity [20]. Therefore, the authors of [19] demonstrated that 96% of households in this city use charcoal, the monthly amount of which was estimated at 800 tons. Consequently, wood is exploited in an unsustainable way in the *miombo* woodland over an increasingly large area [14], leading to deforestation [10,16,21]. This deforestation not only threatens the livelihoods of the population relying on *miombo* woodlands [22] but also undermines DRC's commitment to the Aichi Biodiversity Targets adopted in 2010 [23].

In the rural area of Lubumbashi, the State, which concentrates power and approves all decisions often unilaterally [20], is the main stakeholder in *miombo* woodland management. Its failure to manage the forest has, in recent decades, prompted many Non-Governmental Organizations (NGOs) and other private partners to be involved in forest management. This results in an arena where different stakeholders participate in decision-making on the sustainable management and use of *miombo* woodlands [24]. While participatory forest management considers local communities at the center of all actions [25], the strong power centralization leads to the non-appropriation of this process by local communities in the rural area of Lubumbashi [26,27] due, namely, to the non-inclusion of indigenous knowledge [10,28].

However, in the rural area of Lubumbashi, as elsewhere in the DRC, local communities are rarely implicated in the decision-making process about forest management plans and are generally only encouraged to participate in the implementation phase of these plans. This "up-down" forestry policy leads to misunderstandings and undermines the initiatives of NGOs and public services. In addition, this system has already shown its inadequacies in several developing countries, particularly in Africa [29]. Nevertheless, the participation of local communities in the initial phases of identifying problems about forest resources and in defining the axes of sustainable forest management programs in accordance with their perception (bottom-up) would reverse the trend. It would motivate these local communities to participate in such programs. Yet, this perception of local communities on environmental problems and the availability of forest resources would be a prerequisite to better guide management and restoration measures [30–32].

Several surveys have already been conducted on the perception of environmental problems and the participation of local communities in forest management in Africa [33–35],

in the DRC [36–38], and in the miombo ecoregion [10,39,40]. However, these works have addressed without further exploration the question leading to understanding why local communities have low ownership of forest management programs. In addition, local communities' perception of environmental issues and their participation in *miombo* woodland management remains largely uninformed in the Lubumbashi rural area [14,41]. The present survey highlights the link between the incorporation of local communities' environmental perceptions in the forest management plans, their participation in the activities, and their ownership of the sustainable management programs in the Lubumbashi rural area.

Therefore, this study characterizes local communities' perceptions of *miombo* woodland management in the rural area of Lubumbashi. It tests the hypothesis that (i) local communities perceive deforestation as one of the main environmental problems due to their high level of dependence on forest resources. This perception varies according to the villages covered by the present study; (ii) regardless of the activity, older people with a high level of education would perceive environmental problems more accurately because of their repeated contact with the forest and clear understanding of local contexts; (iii) due to the lack of concerted and inclusive planning, the actions of provincial public services and NGOs would be organized with little participation of local communities, particularly women and young people with lower levels of education, because of their social exclusion and own resignation. The investigations were undertaken in the Lubumbashi rural area, extending over a radius of ±80 km around this city, for a period between 2000 and 2022.

## 2. Materials and Methods

### 2.1. Study Area

This study was conducted in the rural area of Lubumbashi (Figure 1), in the Haut-Katanga province, in the southeastern DRC.

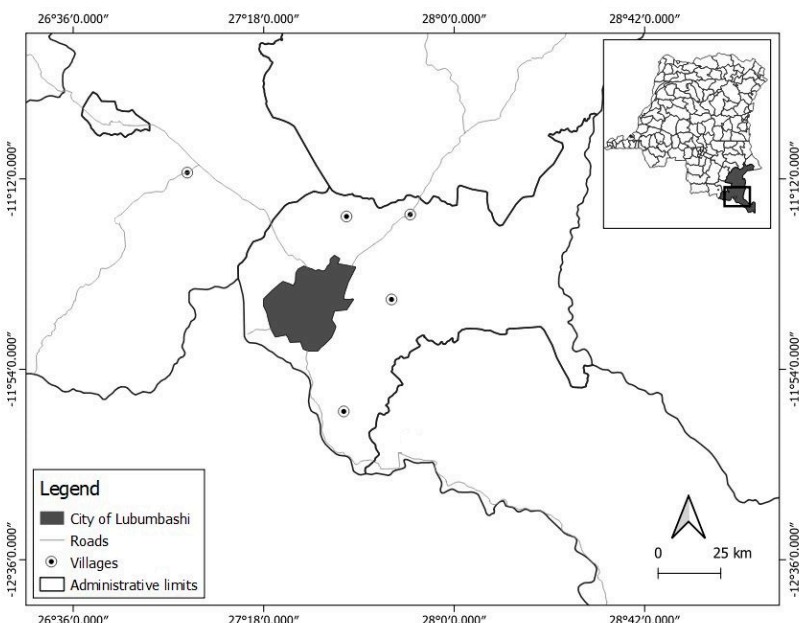

**Figure 1.** Location of Lubumbashi (the gray polygon) and its rural area (the white area around Lubumbashi) in Haut-Katanga province, D.R. Congo. The black dots surrounded by a halo refer to the villages covered by this study within a radius of 80 km from the city of Lubumbashi.

The area has a Cw type climate, according to Köppen's classification, characterized by the alternation of a rainy season (November to March) and a dry season (May to September), separated by two transition months (April and October) [14,42]. While the average annual temperature in the second half of the 20th century has been about 20 °C, ongoing warming has been documented [42]. Total annual precipitation is nearly 1270 mm, with a steadily decreasing number of rainy days since the early 2000s [43]. *Miombo* woodlands,

generally established on ferralsols [44,45], are the pristine vegetation of the rural area of Lubumbashi [14], where they are undergoing fragmentation [46]. The population in the Lubumbashi rural area is predominantly poor, living on less than USD 1.25 [47] and having subsistence agriculture, charcoal production, (illegal) wood exploitation, small (informal) trade, and artisanal mining as their survival activities [13,48,49].

*2.2. Methods*

2.2.1. Village Selection and Sampling

Five villages (Lwisha, Maksem, Mwawa, Nsela, and Texas) were selected within an 80-km radius of Lubumbashi. Following a pre-survey of the main charcoal storage sites in the Lubumbashi (peri-)urban area, these villages are among the most cited charcoal production sites. In addition, these villages are accessible and connected to other surrounding villages. Due to the lack of a population census for more than five decades, there are no official and recent population statistics in DRC, particularly in Lubumbashi rural area. Thus, the surveyed sample was calculated from the total number of households constituting the different villages (Table 1). In these villages, the sample size of the selected households was determined via the Raosoft.com tool, using the Bernoulli sampling equation [38] (Equation (1)). A total of 945 households in the five villages were selected. It should be noted that the household refers to all the occupants of a single dwelling, without these people necessarily being related to each other [50].

$$n = Z^2 \times p \times (1 - p) \times N / Z^2 \times p \times (1 - p) + (N - 1) \times ET^2, \tag{1}$$

where $n$ is the required sample size (calculated separately for each village); $Z$ is the confidence interval ($Z = 1.96$ for $\alpha = 0.05$); $p$ is the estimated proportion of the population with the characteristic of interest (the value of 50% was chosen for this study); $N$ is the population size (the number of households in this study); and $ET$ is the acceptable margin of error.

**Table 1.** Geographic location, demographics, sample size, and structures (NGOs and provincial public services) involved in forest management within the selected villages.

| Villages | Geographical Coordinates * | Total Number of Households | Sample | NGOs | Provincial Services |
|---|---|---|---|---|---|
| Lwisha | 11°10′ S; 27°01′ E | 2670 | 336 | a, b | d, e, g |
| Maksem | 11°19′ S; 27°50′ E | 356 | 186 | - | d, e, f, g |
| Mwawa | 12°03′ S; 27°35′ E | 163 | 115 | a, b, c | d |
| Nsela | 11°20′ S; 27°36′ E | 192 | 129 | - | d, e, f |
| Texas | 11°39′ S; 27°46′ E | 333 | 179 | a | d |
| Total | | 3714 | 945 | | |

* The geographic coordinates of the villages presented in this table were taken from the traditional rulers' houses. *NGOs*: Non-Governmental Organizations; *a*: APRONAPAKAT (Action for the Protection of Nature and Indigenous Peoples of Katanga); *b*: BUCODED (Sustainable Development Consulting Office); *c*: VPPEE (Vision for the Protection of the Environment and the Ecosystem); *d*: Provincial Environmental Coordination, Ministry of Environment and Sustainable Development; *e*: National Forestry Fund Agency, Ministry of Environment and Sustainable Development; *f*: Provincial Division of Agriculture, Ministry of Agriculture; *g*: Provincial Division of Energy, Ministry of Energy and Water Resources; -: lack of NGOs/public services in the villages.

2.2.2. Data Collection

Preliminary investigations consisted of a literature search on environmental issues within the Katangese Copperbelt between the years 2000 and 2022 [14,51–53]. This literature review was conducted through search engines (Scopus, ResearchGate, google scholar, DOAJ) using keywords (Katangese Copperbelt, Lubumbashi, *miombo* woodland, environmental problems, deforestation, trace metal elements, pollution) and Boolean operators [54]. Subsequently, the interviews, guided by semi-structured questionnaires (Provide as Supplementary Material), were conducted according to the objectives of the survey [55], targeting a total of 945 randomly selected and consenting households [56]. Data on the

socio-demographic profile of the respondents (gender, age, stay length in the villages, level of education, professional activities), providing information on both the characteristics of the individuals surveyed and their ability to provide convincing answers to the questionnaire, were collected. The respondents were grouped into three age brackets (Young: 18–35 years old, Adult: 36–60, and Old: ≥61) and stay length in villages (0–5 years, 6–10 years, and >10 years) [57]. For the level of education, respondents were classified into 4 groups: uneducated (did not study), primary school level (had primary education), secondary school level (had secondary education), and university level (had university education). Questions were designed to collect data on respondents' perceptions of environmental problems, the management of *miombo* woodlands, and local communities' involvement in this management. This information on the activities of NGOs and provincial public services was collected from local communities to compensate for the low availability of information from NGOs and provincial public services. All these data were collected using the Kobotoolbox V1.30.1 application. Environmental issues refer to a fact having a negative impact on natural resources and on the global environment, while forest management consists in actions carried out by the provincial public services and the NGOs for the sustainability of the forest resources. Finally, focus groups followed by direct observations were organized in the 5 villages with several participants, whose numbers varied between 9 and 12 [58] to verify the veracity of the information from individual interviews and to inventory the activities carried out by provincial public services and NGOs in favor of forest management. Triangulation of information with the literature review results was also done to verify the veracity of the information from individual interviews. All these surveys were conducted between 13 February and 15 March 2022.

2.2.3. Data Analysis

The qualitative data were analyzed using descriptive statistics (relative frequencies). To highlight the difference between the proportion of environmental issues perceived by local communities, a non-parametric Friedman test was conducted. This test allows several matched samples, to determine if the values of the variables are significantly different [59]. The post-hoc pairwise comparison test was conducted to prioritize villages and perceived environmental problems. A statistical significance level of 0.05 was used for these comparisons [60]. These inferential tests were conducted using the rstatix package of R 4.1.0 software. To establish the relationship between the perception of the heads of households and the elements of their socio-demographic profile, a multiple correspondence factor analysis (MCA) [31] was performed. In addition, this factor analysis was performed to highlight the profile of respondents participating in the activities of NGOs/provincial public services. The proximity of the variables indicates their correspondence. This factor analysis was conducted using JMP 15.2 PRO software. The Jaccard Index [61] (Equation (2)) was calculated, using Past version 4.05 software, to highlight the similarity between the environmental problems identified in the literature according to the two approaches used (Empirical Research and Individual Interviews). On the other hand, this index was calculated to highlight the level of similarity between the environmental problems identified in the literature, those perceived by the respondents, and the activities of the NGOs/provincial public services.

$$J = a/a + b + c, \tag{2}$$

where $J$ is the Jaccard index; and in the case of this study, $a$ is the number of problems covered by the actions of the two NGOs and/or provincial public services; $b$ and $c$ are the number of issues covered by the activities of one of the two NGOs and/or provincial public services, concerned.

## 3. Results

### 3.1. Sociodemographic Profile of Heads of Households

Most of the respondents were men in terms of age categories, and more than 90% were young and adult. In addition, nearly $\frac{3}{4}$ of the heads of households live in the villages for

10 years or less, with a low level of education (uneducated or primary school). Agriculture, combined with charcoal production, is the main source of income for most of the households surveyed. In terms of variability of respondents across villages, Lwisha and Nsela villages recorded many women, while older respondents (≥61 years old) were least represented in all villages. The different villages were dominated by respondents which spent less than 5 years in the village. The respondents with a university level are only represented in Lwisha, Mwawa, and Nsela (Table 2).

**Table 2.** Socio-demographic characteristics of heads of households in different villages in the rural area of Lubumbashi.

| Elements of the Socio-Demographic Profile | Villages | | | | | Average Percentage |
|---|---|---|---|---|---|---|
| | Lwisha * *n* = 336 | Maksem *n* = 186 | Mwawa *n* = 115 | Nsela *n* = 129 | Texas *n* = 179 | |
| Gender (%) | | | | | | |
| Woman | 50.9 | 37.6 | 47.8 | 52.7 | 44.1 | 46.9 |
| Male | 49.1 | 62.4 | 52.2 | 47.3 | 55.9 | 53.1 |
| Age range (%) | | | | | | |
| Young | 33.6 | 57.5 | 48.7 | 47.3 | 36.9 | 42.7 |
| Adult | 58.1 | 40.9 | 47.0 | 48.1 | 58.1 | 52.0 |
| Old | 8.3 | 1.6 | 4.4 | 4.7 | 5.0 | 5.4 |
| Time spent in the villages (%) | | | | | | |
| 0–5 years | 44.4 | 48.9 | 34.8 | 57.4 | 42.5 | 45.6 |
| 5–10 years | 26.2 | 21.5 | 19.1 | 31.0 | 21.8 | 23.9 |
| >10 years | 29.5 | 29.6 | 46.1 | 11.6 | 35.8 | 30.5 |
| Level of education (%) | | | | | | |
| Uneducated | 14.3 | 24.2 | 19.1 | 37.2 | 26.3 | 24.2 |
| Primary school | 35.4 | 39.3 | 47.8 | 30.2 | 43.0 | 39.1 |
| Secondary school | 47.9 | 36.7 | 29.6 | 31.0 | 30.7 | 35.2 |
| University level | 2.4 | 0.0 | 3.5 | 1.6 | 0.0 | 1.5 |
| Main activities (%) | | | | | | |
| Cultivator | 92.0 | 92.5 | 93.9 | 78.3 | 97.8 | 91.5 |
| Charcoal producer | 5.4 | 6.5 | 3.5 | 19.4 | 0.6 | 6.4 |
| Art Sculptor | 2.4 | 1.1 | 0.0 | 1.6 | 1.1 | 1.5 |
| Logger of timber | 0.0 | 0.0 | 2.6 | 0.8 | 0.0 | 0.4 |
| NTFP Collector | 0.3 | 0.0 | 0.0 | 0.0 | 0.6 | 0.2 |

* *n*: sample size; *NTFP*: non-timber forest products.

### 3.2. Main Environmental Problems as Found in the Literature and Perceived by the Local Communities

3.2.1. Literature Review from 2000 to 2022 in the Katangan Copperbelt

The bibliographic review identified 120 documents written based on empirical research (including 90 scientific publications, 15 book chapters, and 16 reports) and 35 written through individual interviews (including 25 peer-reviewed articles, four book chapters, and three reports). This total of 155 documents is distributed among 121 different authors, 99 of whom conducted empirical research. Regardless of the research approach, this literature review notes that most publications within the Katangan Copperbelt focus on environmental issues related to deforestation and forest degradation, as well as trace metal pollution of soils and water. Air pollution is the least researched issue by empirical investigations, while biological erosion and invasion are the least addressed problems in interview-based surveys (Table 3). Jaccard similarity between the problems identified by the two research approaches is 87.5%.

**Table 3.** Environmental problems identified by the literature review on the Katangan Copperbelt.

| Key Issues (%) | Approaches Used | |
|---|---|---|
| | Empirical Research * $n$ = 120 | Individual Interviews $n$ = 35 |
| Deforestation and forest degradation | 44.2 | 37.1 |
| Trace metals in soil | 28.3 | 20.0 |
| Water pollution | 8.3 | 14.3 |
| Climate change | 6.7 | 5.7 |
| Soil fertility loss | 5.8 | 8.6 |
| Biological erosion | 5.0 | 2.9 |
| Air pollution | 1.7 | 8.6 |
| Biological invasion | - | 2.9 |
| Total | 100.0 | 100.0 |

* $n$: sample size; -: no publication found.

### 3.2.2. Perception of Environmental Problems by Local Communities in the Rural Area of Lubumbashi

More than half of the respondents have identified the problems of soil fertility loss and deforestation as the main environmental problems in their area. On the other hand, climate change was identified by only about 15% of the heads of households surveyed, compared to nearly 5% for water pollution. The Friedman statistical test reveals significant differences in perceived problems ($p < 0.05$). The pairwise comparison shows that soil fertility loss and deforestation are perceived similarly ($p > 0.05$) and more than the rest of the problems. Similarly, this comparison shows that the level of perception of environmental problems remains similar between villages ($p < 0.05$; Table 4).

**Table 4.** Main environmental problems perceived in the different villages of the Lubumbashi rural area.

| Environmental Problems (%) * | Villages | | | | | Signif. |
|---|---|---|---|---|---|---|
| | Lwisha $n$ = 336 | Maksem $n$ = 186 | Mwawa $n$ = 115 | Nsela $n$ = 129 | Texas $n$ = 179 | |
| Soil fertility loss | 43.2 | 21.0 | 55.7 | 47.3 | 43.0 | a |
| Deforestation | 22.9 | 34.4 | 8.7 | 13.2 | 31.3 | a |
| Climate change | 6.9 | 15.1 | 13.1 | 18.6 | 10.1 | ab |
| Water pollution | 7.7 | 2.7 | 0.0 | 5.4 | 0.6 | b |
| Signif. | a | a | a | a | a | - |

* The sum of the frequencies does not reach 100% because the category "no perceived problem" has been removed from the table. $n$: sample size; Signif.: letters indicating significant differences between perceived environmental problems and between villages.

### 3.3. Sociodemographic Determinants of Local Communities' Perception of Environmental Problems

Soil fertility loss was cited mostly by adult women, who often grow crops, with a low level of education (uneducated and primary school level). Deforestation was mentioned more by adult men with a secondary level of education. In addition, respondents with a university level, charcoal producers, timber harvesters, NTFP collectors, and sculptors have shown a tendency to have a less focused perception of environmental problems than other categories of respondents. Finally, young respondents and those who have been in the villages for less than 10 years summarize their perception of environmental problems only in terms of climate change, but the majority did not mention any environmental problem (Figure 2).

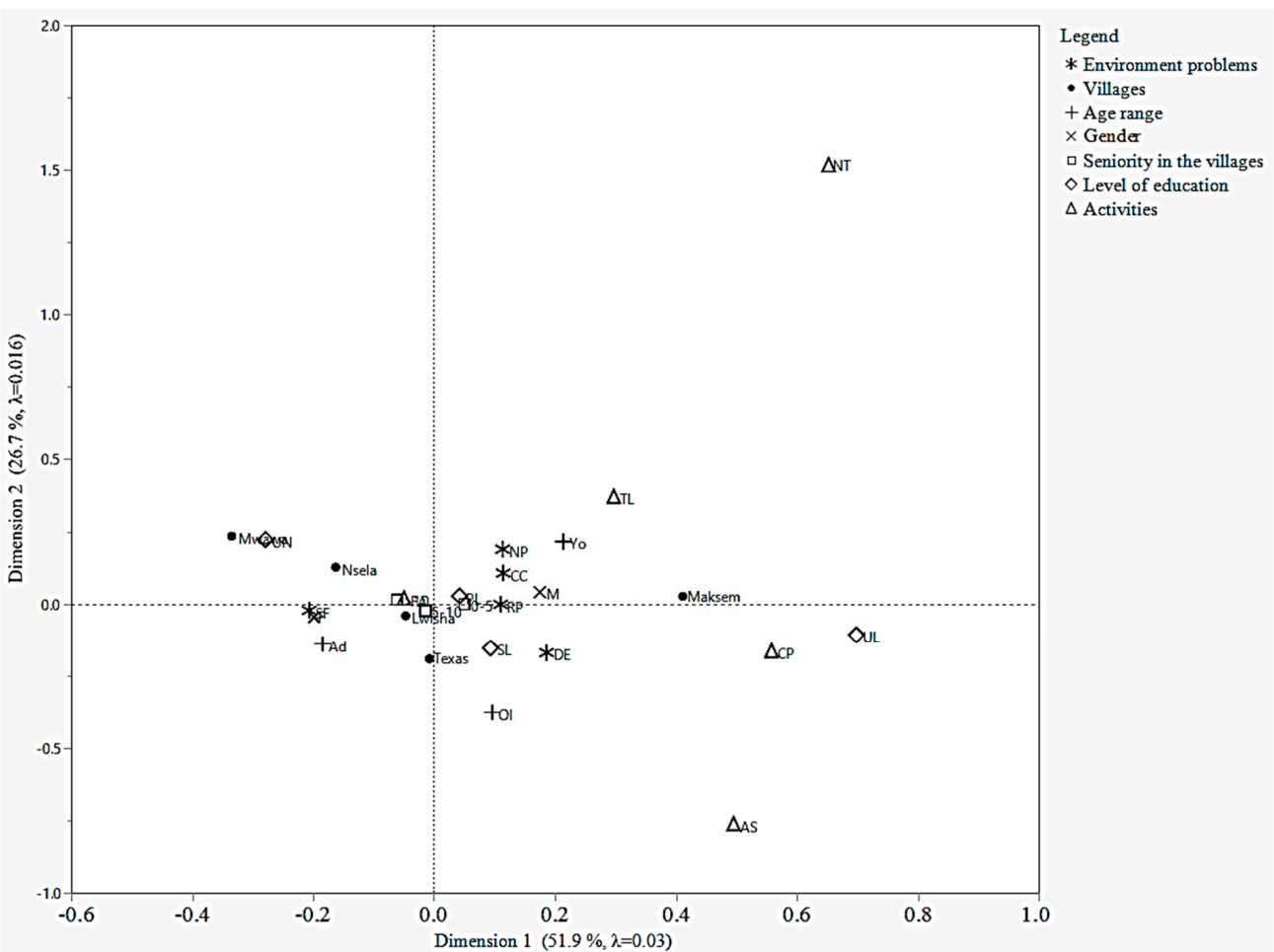

**Figure 2.** Determinants of local communities' perception of environmental problems in the different villages of the rural area of Lubumbashi, depicted in a plan consisting of the first two dimensions summarizing 78.6% of the information following a multiple correspondence factor analysis. *F*: Female; *M*: Male; *Yo*: Young (18–35 years); *Ad*: Adult (36–60 years); *Ol*: Old (61 years and older); 0–5, 5–10, >10: seniority between 0–5 years, 5–10 years, 10 years and older, respectively; *UN*: Uneducated; *PL*: primary school level; *SL*: secondary school level; *UL*: University level; *FA*: Farmer; *CP*: Charcoal producer; *TL*: Timber logger; *NT*: NTFP collector; *AS*: Art sculptor; *SF*: Soil fertility loss; *DE*: Deforestation; *CC*: Climate change; *RP*: Water pollution; *NP*: No perceived environmental problem.

*3.4. Perceived Actions of NGOs/Provincial Public Services and Participation of Local Communities*

3.4.1. Actions of ONGs and Provincial Public Services as Perceived by Local Communities

According to the respondents, the actions of the NGOs can be summarized as training and sensitization of local communities for the sustainable exploitation of *miombo* woodlands, distribution of seedlings (e.g., *Acacia auriculiformis* A.Cunn. ex Benth., *Psidium guajava* L., *Persea americana* Mill., *Moringa oleifera* Lam., *Pinus* sp., and *Citrus* spp.) for homestead gardens, as well as reforestation activities. Specifically, nearly $\frac{3}{4}$ of respondents noted the actions of the NGO APRONAPAKAT in terms of reforestation, collection of seeds (of tree species such as *Afzelia quanzensis* Welw., *Albizia adianthifolia* (Schumach.) W.F.Wight, *Albizia antunesiana* Harms, *Brachystegia spiciformis* Benth, *Julbernardia paniculata* (Benth.) Troupin, *Julbernardia globuflora* (Benth.) Troupin) and the implantation of nurseries. Eighteen hundred *Albizia lebbeck* seedlings and nearly seven hundred and fifty *Leucaena leucocephala* seedlings were found in a nursery in preparation for reforestation in the Mwawa forest concession. Nearly 20% of respondents claimed that this NGO had distributed seedlings of *Acacia auriculiformis* A.Cunn. ex Benth. for the hut garden in Texas village. Regarding the NGOs BUCODED and VPPEE, more than 80% of respondents perceive actions related

to the drafting of simple management plans for the benefit of the Forest Concessions of the Local Communities of Lwisha and Mwawa, respectively. However, the local communities perceive that the actions of provincial public services are limited to training and sensitization of local communities on the sustainable exploitation of *miombo* woodland and the collection of taxes. More than $\frac{3}{4}$ of the respondents confirmed that the Environmental services and the National Forestry Fund carry out their actions in training and sensitization on sustainable exploitation, tax collection, and distribution of seedlings (*A. auriculiformis* and *Pinus* sp.) to the communities. The Environment and Energy services are involved in forest control and tax collection, respectively, while the agriculture department is involved only in training and sensitization of local communities (Table 5).

**Table 5.** Actions of NGOs and provincial public services, as perceived by local communities in the different villages of the rural area of Lubumbashi.

| Actions Taken | Citation Frequencies (%) * | | | | | | |
|---|---|---|---|---|---|---|---|
| | NGOs | | | Provincial Public Services | | | |
| | APRONAPAKAT $n = 630$ | BUCODED $n = 451$ | VPPEE $n = 115$ | Envir. $n = 945$ | FFN $n = 651$ | Agri. $n = 315$ | Energ. $n = 522$ |
| Forest control | - | - | - | 14.7 | - | - | - |
| Distribution of seedlings | 17.1 | - | - | 4.8 | 33.3 | - | - |
| Elaboration of the SMP | - | 28.4 | 41.7 | - | - | - | - |
| Training and sensitization | 8.6 | 58.5 | 44.3 | 2.9 | 13.5 | 46.3 | - |
| Installation of nurseries | 6.2 | - | - | - | - | - | - |
| Tax collection | - | - | - | 76.9 | 53.2 | - | 92.1 |
| Reforestation | 38.8 | - | - | - | - | - | - |
| Seeds harvesting | 24.2 | - | - | - | - | - | - |

* The values correspond to the proportions (%) of heads of households surveyed who perceived the actions. The proportions of respondents who did not perceive the actions of NGOs and provincial public services are not presented in this table. *APRONAPAKAT*: Action for the Protection of Nature and Indigenous Peoples of Katanga; *BUCODED*: Sustainable Development Consulting Office; *VPPEE*: Vision for the Protection of the Environment and the Ecosystem; *Envir.*: Provincial Coordination of Environment, Ministry of Environment and Sustainable Development; *FFN*: National Forestry Fund Agency, Ministry of Environment and Sustainable Development; *Agri.*: Provincial Division of Agriculture, Ministry of Agriculture; *Energ.*: Provincial Division of Energy, Ministry of Energy and Hydraulic Resources; *SMP*: Simple Management Plan; *n*: sample size, calculated according to the villages of intervention of the provincial public services and the NGOs; -: lack of action by relevant NGOs/public services.

### 3.4.2. Similarity between the Actions of NGOs/Provincial Public Services, the Problems Identified in the Literature, and the Perception of Local Communities

Local communities' perceptions of environmental issues are about half as similar as those identified in the literature review for the Katangan Copperbelt. The overall actions of NGOs and provincial public services remain almost similar, addressing only a small proportion of the issues identified in the literature review and by the respondents. Moreover, these actions remain related to deforestation, climate change, and biological erosion, as detailed in Table 6.

**Table 6.** Suitability of the actions of the NGOs/provincial public services to the problems identified in the literature and by the local communities.

| Problems Identified in the Literature * | Problems Cited by Local Communities | NGOs | | | Provincial Public Services | | | |
|---|---|---|---|---|---|---|---|---|
| | | APRONAPAKAT | BUCODED | VPPEE | ENV. | FFN | AGRI | ENERG. |
| Deforestation and forest degradation | Deforestation | 2, 4, 5, 6, 7 | 3, 4 | 3, 4 | 1, 2, 4 | 2, 4 | - | - |
| Trace metals in soil | - | - | - | - | - | - | - | - |
| Water pollution | Water pollution | - | - | - | - | - | - | - |
| Climate change | Climate change | 2, 4 | 3, 4 | 3, 4 | 1, 2, 4 | 2, 4 | - | - |
| Soil fertility loss | Soil fertility loss | - | - | - | - | - | 4 | - |
| Biological erosion | - | 2, 4 | 3, 4 | 3, 4 | 1, 2, 4 | 2, 4 | - | - |
| Air pollution | - | 2 | - | - | 2 | 2 | - | - |
| Biological invasion | - | - | - | - | - | - | - | - |
| - | - | - | - | - | 8 | 8 | - | 8 |

\* The values in the table represent the actions of the NGOs/public services. 1: Forest control; 2: Distribution of seedlings; 3: Elaboration of Simple Management Plans; 4: Training and sensitization; 5: Installation of nurseries; 6: Reforestation; 7: Seed harvesting; 8: Tax collection. Blank spaces in the table indicate no actions. *NGOs*: Non-Governmental Organizations; *APRONAPAKAT*: Action for the Protection of Nature and Indigenous Peoples of Katanga; *BUCODED*: Sustainable Development Consulting Office; *VPPEE*: Vision for the Protection of the Environment and the Ecosystem; *Envir.*: Provincial Coordination of the Environment, Ministry of Environment and Sustainable Development; *FFN*: National Forestry Fund, Ministry of Environment and Sustainable Development; *Agri.*: Provincial Division of Agriculture, Ministry of Agriculture; *Energ.*: Provincial Division of Energy, Ministry of Energy and Hydraulic Resources; -: lack of actions or environmental problems corresponding to the perception of Local communities or NGOs/provincial public services actions.

While there is a similarity of more than 75% between the areas of action of the different NGOs carrying out environmental actions, the different actions of the provincial public services are only about 20% similar to each other. On the other hand, the actions of the NGOs are almost 80% similar to those of the provincial public environmental service and the National Forestry Fund. Although the environmental problems perceived by local people only meet those in the literature at 50%, the areas of action of the NGOs and the provincial public services only cover less than 50% of the environmental problems identified in the literature. Finally, the results show that the actions of NGOs and provincial public services only weakly (40%) meet the problems identified by local communities (Table 7).

**Table 7.** Jaccard similarity index calculated between the literature, the perception of local communities, and the actions of NGOs and provincial public services.

| | Litt. * | APRONAPAKAT | BUCODED | VPPEE | ENV. | FFN | AGRI | ENERG. |
|---|---|---|---|---|---|---|---|---|
| APRONAPAKAT | 0.50 | | | | | | | |
| BUCODED | 0.38 | 0.75 | | | | | | |
| VPPEE | 0.38 | 0.75 | 1.00 | | | | | |
| Envir. | 0.44 | 0.80 | 0.60 | 0.60 | | | | |
| FFN | 0.44 | 0.80 | 0.60 | 0.60 | 1.00 | | | |
| Agri | 0.13 | 0.00 | 0.00 | 0.00 | 0.00 | 0.00 | | |
| Energ. | 0.00 | 0.00 | 0.00 | 0.00 | 0.20 | 0.20 | 0.00 | |
| LC | 0.50 | 0.33 | 0.40 | 0.40 | 0.29 | 0.29 | 0.25 | 0.00 |

\* *Litt.*: Literature; *APRONAPAKAT*: Action for the Protection of Nature and Indigenous Peoples of Katanga; *BUCODED*: Sustainable Development Consulting Office; *VPPEE*: Vision for the Protection of the Environment and the Ecosystem; *Envir.*: Provincial Coordination of Environment, Ministry of Environment and Sustainable Development; *FFN*: National Forestry Fund Agency, Ministry of Environment and Sustainable Development; *Agri.*: Provincial Division of Agriculture, Ministry of Agriculture; *Energ.*: Provincial Division of Energy, Ministry of Energy and Hydraulic Resources; *SMP*: Simple Management Plan; *n*: sample size, calculated according to the villages of intervention of the provincial public services and the NGOs; *LC*: Local Communities.

### 3.4.3. Participation of Local Communities in the Actions of NGOs and Provincial Public Services

The participation of local community members in the activities of provincial public services and NGOs does not exceed $\frac{1}{4}$ of the respondents surveyed, with slightly high participation in the actions of NGOs. In the case of APRONAPAKAT, most respondents

stated that they had participated in the distribution of seedlings, the harvesting of seeds, and the installation of nurseries. The participation of respondents in the actions of the NGOs BUCODED and VPPEE is noted during activities related to the development of the simple management plan. Regarding the provincial public services, respondents stated that they have participated more in activities related to training and sensitization among local communities and the distribution of seedlings. The low level of participation in the activities of the provincial public services and NGOs is justified by the unavailability of the respondents surveyed or their exclusion by these structures. In particular, the low participation of local communities in the activities of the provincial public services is justified mainly by a lack of interest (Table 8).

**Table 8.** Participation of the local communities in the actions carried out by the NGOs and the provincial public services, in the different villages of the rural area of Lubumbashi.

| Actions Taken * | NGOs | | | Provincial Public Services | | | |
|---|---|---|---|---|---|---|---|
| | APRONAPAKAT $n = 630$ | BUCODED $n = 451$ | VPPEE $n = 115$ | ENV. $n = 945$ | FFN $n = 651$ | AGRI. $n = 315$ | ENERG. $n = 522$ |
| Participation of local communities in the actions (%) | | | | | | | |
| Forest control | - | - | - | 0.0 | - | - | - |
| Seedling distribution | 7.8 | - | - | - | 7.2 | - | - |
| Elaboration of the GMP | - | 14.9 | 21.7 | - | - | - | - |
| Training and sensitization | 1.2 | 7.0 | 6.2 | 1.9 | 2.8 | 4.8 | - |
| Setting up of the plant nurseries | 5.1 | - | - | - | - | - | - |
| Tax collection | - | - | - | 0.0 | 0.0 | - | 0.0 |
| Reforestation | 4.4 | - | - | - | - | - | - |
| Harvesting the seeds | 6.0 | - | - | - | - | - | - |
| Factors explaining the low participation of some respondents in the actions (%) | | | | | | | |
| Lack of information | 8.4 | 7.1 | 5.2 | 3.7 | - | - | - |
| Lack of special interest | 9.3 | 16.2 | 15.7 | - | - | 29.8 | - |
| Exclusion in actions | 13.5 | 18.4 | 10.4 | 11.0 | 9.7 | - | - |
| No empowerment | - | - | - | 74.6 | 71.3 | - | 74.9 |
| Little availability | 35.7 | 33.5 | 40.9 | - | - | 83.2 | 19.2 |
| Loss of time | 14.2 | 16.0 | 13.9 | 9.6 | 10.8 | 35.9 | 13.8 |

* The values correspond to the proportion (%) of respondents (not) involved in these activities. *APRONAPAKAT*: Action for the Protection of Nature and Indigenous Peoples of Katanga; *BUCODED*: Sustainable Development Consulting Office; *VPPEE*: Vision for the Protection of the Environment and the Ecosystem; *ENV*.: Provincial Coordination of the Environment, Ministry of the Environment and Sustainable Development; *FFN*: National Forestry Fund, Ministry of the Environment and Sustainable Development; *AGRI*.: Provincial Division of Agriculture, Ministry of Agriculture; *SMP*: Simple Management Plan; *ENERG*.: Provincial Division of Energy, Ministry of Energy and Hydraulic Resources; *n*: sample size, calculated according to the villages of intervention of the NGOs and provincial public services; **-**: no action and no participation in the actions of the corresponding NGOs/provincial public services.

The various activities organized by the NGOs/provincial public services are more likely to be attended by adult and aged men, with secondary school and university levels, who have been in the village for more than 10 years, and whose main activities are subsistence farming, charcoal production, logging for timber and art sculpting. On the other hand, women, and youth with low levels of education (uneducated and primary school level), less than 10 years spent in the village, and NTFP collectors have low participation in these activities (Figure 3).

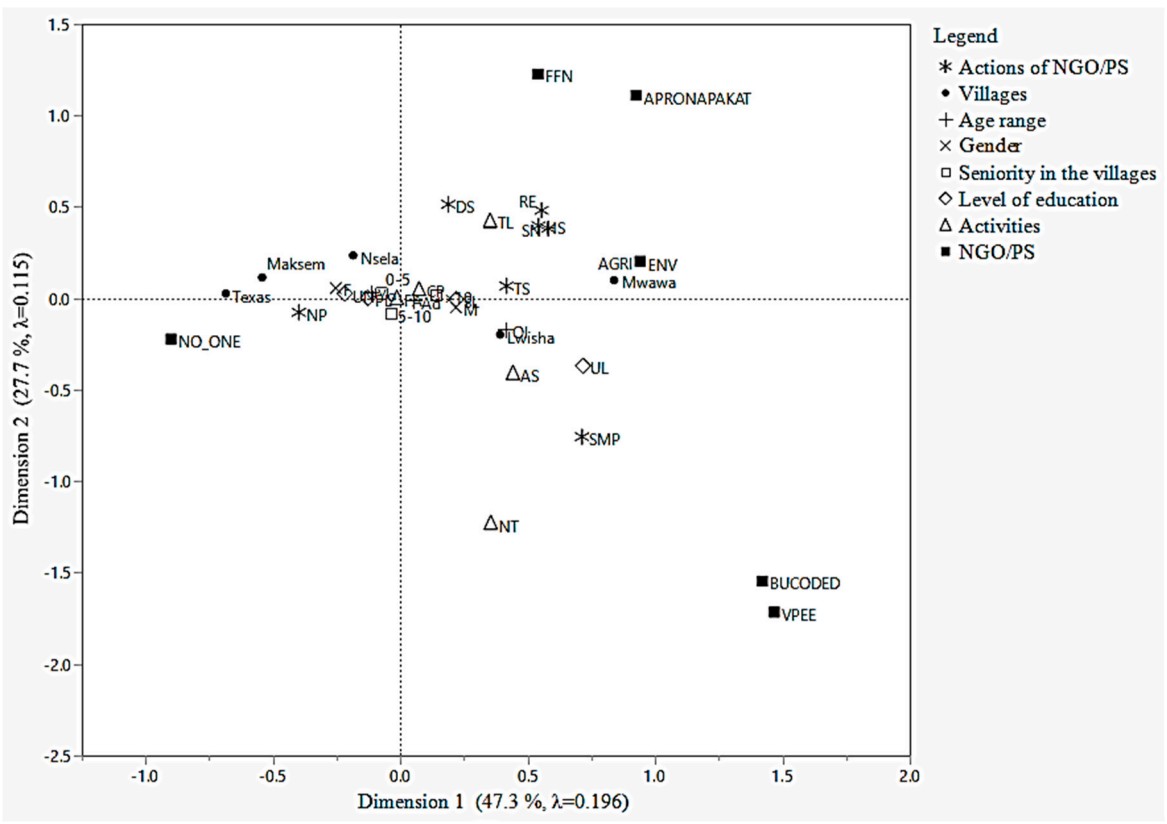

**Figure 3.** Sociodemographic profile of the actors participating in the different activities organized by the NGOs/provincial public services (*PS*), in a plan made up of the first two dimensions summarizing 75% of the information following a multiple correspondences factorial analysis. *F*: Female; *M*: Male; *Yo*: Young (18–35 years old); *Ad*: Adult (36–60 years old); *Ol*: Old (61 years and older); 0–5, 5–10, >10: seniority between 0–5 years, 5–10 years, 10 years and older, respectively; *UN*: Uneducated; *PL*: primary school level; *SL*: secondary school level; *UL*: university level; *FA*: Farmer; *CP*: Charcoal producer; *TL*: Timber logger; *NT*: NTFP collector; *AS*: Art sculptor; *APRONAPAKAT*: Action for the Protection of Nature and Indigenous Peoples of Katanga; *BUCODED*: Sustainable Development Consulting Office; *VPPEE*: Vision for the Protection of the Environment and the Ecosystem; *ENV.*: Provincial Coordination of the Environment, Ministry of the Environment and Sustainable Development; *FFN*: National Forestry Fund Agency, Ministry of the Environment and Sustainable Development; *AGRI.*: Provincial Division of Agriculture, Ministry of Agriculture; *ENERG.*: Provincial Division of Energy, Ministry of Energy and Hydraulic Resources; DS: Distribution of seedlings for hut gardens; *TS*: Training and sensitization; *SN*: Installation and maintaining the nursery; *NP*: Not participating; *SMP*: Activities related to the simple management plan; *RE*: Reforestation; *HS*: Harvesting of seeds of native forest species.

## 4. Discussion

### 4.1. Methods Used in This Survey and Limitations of the Findings

The semi-structured interview was conducted in this study using Bernoulli sampling. However, there are several interview techniques (structured and unstructured) and sampling methods that can be used in the social sciences: random area limitations sampling, strata sampling, and quota sampling [62]. The different combinations of these interview techniques and sampling methods would not necessarily produce results similar to the present study, despite using them on the same survey questionnaire and with the same individuals. While the choice of the technique and sampling method used in the present study would be justified appropriately by letting the interviewer guide the interview according to the objectives set (avoiding response slippage and unspoken words) but still leaving the latitude for the interviewee to express themselves freely [63] and to remain empirical and

logical in sampling, the use of other interview techniques and sampling methods requires detailed knowledge of the population from which the sample would be tired [56].

Furthermore, the numbers of respondents per village are more than sufficient to generalize the results to the sampled populations. Indeed, the authors of [56,64] propose a sample size of 30–40 respondents that would be sufficient to generate acceptable statistics, particularly in the absence of reliable demographic statistics that could be used to tire out the population sample. Nevertheless, despite the focus groups and triangulations conducted during the present study to validate the results, further triangulations between the responses of the heads of households surveyed and those of the provincial services and NGOs would be an important contribution to the present results. The present study's findings could be generalized to other villages around Lubumbashi since the local communities in the Lubumbashi rural area are almost ethnically the same and have nearly the same socio-demographic profile. Furthermore, the applicability of the results and conclusions of this paper in other areas will be determined by the conformity of the conditions related to the state of the forest resources, the management system, and the socio-demographic profile of the stakeholders in this study area.

*4.2. Environmental Issues in the Rural Lubumbashi: Hierarchy, Magnitude, and Veracity of Information Sources*

Deforestation and pollution by trace metals are the two major environmental issues identified in the literature. Pollution results from mining activities and is responsible for environmental degradation, especially habitat fragmentation in the rural area of Lubumbashi [65,66] and in Zambia [12]. In addition, the literature has highlighted a gradual disappearance of woody species on soils containing high concentrations of trace metal elements, which may also justify tree cover loss. Mining activities thus lead to deforestation in an (in)direct way, justifying the feedback of two processes, as perceived by the population. Furthermore, perceived by local communities as a major environmental problem independently of the villages, the soil fertility loss because of agriculture-driven deforestation [67] or charcoal production [19], is justified by the disruption of the soil's supply of mineral elements by the litter of *miombo* woodland species [68]. For this reason, it is recognized that one of the most critical anthropogenic impacts in the study area is found to be the removal of the original vegetation cover, accompanied by its replacement by another land use category [16,49,69,70]. Ref. [46] also pointed out that the significantly accelerated deforestation in recent decades is a consequence of, among other things, the high demand for dendroenergy. Yet, for local communities living under otherwise deleterious economic conditions, charcoal production constitutes an opportunity to increase income through the exploitation of *miombo* woodland, whose patch abundance is constantly regressing over short distances to the city [13], including around the villages surveyed as part of this study. Indeed, charcoal production allows rural households to make available financial resources complementary to agricultural activities [71], justifying the link between these two activities in the rural area of Lubumbashi.

On the other hand, local communities have reported only half of the environmental issues identified by the literature review. This result is justified by the fact that most local community members have a low level of education, limiting their integration into labor markets, and are involved in shifting cultivation and charcoal production, thus perceiving the related problems more clearly than other environmental issues. Indeed, nearly 70% of RDC's rural population has a low education level [72]. In addition, in the context of a tight labor market, agriculture and the exploitation of forest resources become alternative sources of households' income and subsistence, with little concern for the sustainability of resources [20,73]. The hypothesis that local communities perceive deforestation as one of the main environmental problems is confirmed by this finding.

### 4.3. Determinants of Community Members' Perception of Environmental Problems and Participation in Forest Management Activities

The perception of environmental problems is strongly linked with households' profiles. This is particularly true for gender due to the social division of labor [71,74]. Indeed, the fact that soil fertility loss is perceived by women and deforestation by men corroborates Ref. [74] findings that an individual's gender determines the types of activities to be carried out in society in sub-Saharan Africa. Certainly, men tend to generally engage in work that increasingly requires muscular strength, such as land clearing and/or charcoal production [75,76]. For The influence of age and seniority on the perception of environmental problems is because processes such as deforestation and soil fertility loss require relatively long spatiotemporal observation. Ref. [26] demonstrated that in DRC, local community members' perception of environmental problems increases with age and seniority in the villages. While the perception of deforestation increases with the respondents' education level (mostly men), an opposite trend is observed regarding soil fertility loss. In fact, educated heads of households, who are mostly men, have more knowledge about current environmental issues, thus corroborating the findings of Ref. [50] in DRC. In addition, in Congolese rural areas, women remain marginalized in terms of access to certain rights such as basic education, and are thus dedicated to domestic activities, farming, and NTFP collection [72]. These results confirm that older people with a high level of education would perceive environmental problems more accurately.

### 4.4. Actions of Provincial Public Services and NGOs and Participation of Local Communities

Provincial public services and NGOs are taking a series of actions to promote the sustainable use of *miombo* woodlands. However, their actions remain insufficient regarding the magnitude of the environmental problems identified in the literature review and are of less priority levels for local communities in the rural area of Lubumbashi. Although the choice of areas of action and the implementation of activities by provincial public services and NGOs may be the result of their original missions, the low availability of sufficiently qualified human resources and financial resources, resulting in low organizational capacity [10,77]. Indeed, the Congolese government, the main source of funding for provincial public services, allocates little financial resources of its budget to forestry management (less than 1%) and agriculture (less than 3%). To compensate for this, provincial public services make tax collection their priority [78] to make up for low operating costs [20] and alleviate the underpayment of the personnel. The results of this study converge with those obtained by the authors of [10,79], suggesting that in the *miombo* ecoregion, public services and NGOs only direct their activities according to existing capacities and means. In addition, reforestation programs in the rural area of Lubumbashi use nearly 70% of native tree species, ensuring that the structure, functioning, and composition of the *miombo* woodlands are maintained [80]. However, the 30% of exotic species used by provincial public services and NGOs for reforestation constitute a threat to the native *miombo* trees [81], namely through the alteration of its functions and by inducing biodiversity loss [18,80].

In addition, the results highlighted the low participation of local communities (particularly women and youth with low levels of education) in the activities of NGOs and provincial public services. Indeed, women and youth are the most marginalized classes in some rural societies, such as in rural Lubumbashi [72]. Despite the social division of labor, these classes are subject to prohibitions that can go as far as excluding them from participating in meetings in the same way as older men [72,82]. Furthermore, nearly $\frac{3}{4}$ of the population living in rural areas has a low level of education in DRC [72,83]; their exclusion and lack of availability for activities organized by NGOs and provincial public services would justify the low participation noted by the results of this study. Non-accountability; non-compliance with legal texts; divergence of opinions; exclusion and marginalization of local communities would induce their low participation in the sustainable management of *miombo* woodlands [27]. In fact, local communities are wary of NGOs, they report to be more interested in obtaining short-term results to secure funds from external donors and

to legitimize their position as "spokespersons" for the local community [84]. On the other hand, the weaknesses of inherent to the government and the multiplicity of taxes weighing on the income of poor farmers [85], would favor a negative connotation on provincial public services in the mindset of the surrounding community of *miombo* woodlands. Overall, our results stress the poor participation of local communities in forest management due to the strong centralization of power by provincial public services and NGOs, low consultation, and non-inclusiveness of actors, as per the results of many studies conducted in Central Africa [29,86,87], in other regions of the DRC [78,88] and in the *miombo* ecoregion [10,27,79]. This result confirms that the actions of provincial public services and NGOs would be organized with low participation from local communities.

### 4.5. Implications of the Results for Sustainable Management of miombo Woodlands

Subsistence agriculture and charcoal production are two anthropogenic activities contributing largely to deforestation in rural Lubumbashi [13], one consequence of which is soil fertility loss. Eleanor et al., [89] demonstrated that forest cover, through its litter and symbiotic relationships, improves the physical, chemical, and biological properties of the soil, making it attractive to farmers. In this context, agroforestry is a solution to contribute to restore the fertility of cultivable land by limiting the itinerancy experienced by farmers in the region [90], which will certainly contribute to reduce the anthropic pressure on the forest resources of the region [91]. For example, the Mampu agroforestry system on the Bateke Plateau of DRC can be expected to yield an average of 1.5 tons of charcoal, 1.25 tons of cassava, 70 kg tons of maize, and 0.75 kg of honey, per hectare, which is much higher than the average production of degraded fallows in the region [92]. However, the adoption of this new approach by local communities living in the *miombo* woodland area would be hindered by constraints related to the level of education of local community members (most of whom have a low level of education) and those related to the social distribution of tasks (most of which are carried out by women and children) [74]. Cases of difficulties in adopting new farming practices due to low levels of education and the social division of labor have been documented in Morocco and northwest Benin [93,94]. To do this, awareness-raising regarding the improvement of the level of education and the improvement of the perception of gender should be put in place, with the support of provincial public services and NGOs.

In the rural area of Lubumbashi, community forestry is also emerging as a new opportunity for forest management by local communities on which their survival depends. Indeed, in two case studies in Kongo Central, DRC, local forest management lead by local rulers or lineage leaders has shown its limitations, while the practice of rapid return to forest (for agriculture or charcoal exploitation) hinders the possibility of regeneration of most local species [95]. This trend has also been observed in the rural area of Lubumbashi [20]. It is in this context that community forestry appears as a solution to mitigate deforestation, especially since it is a form of forest management and exploitation in which local communities elaborate together and apply rules of access and use of the forest and participate in its exploitation [96]. However, to make this approach sustainable, it is crucial to support communities in managing and allocating revenues from these products [97].

Furthermore, NGOs and provincial public services should reorient and intensify actions based on the problems of the region, as identified by local communities and literature review, to foster local community participation and ownership. This requires the promotion of concerted and inclusive planning, involving local communities and other stakeholders [38], including the miners. However, in the context of low local community participation in forest management planning and programs, due to the forestry policy inadequacy and poor accountability [9,10], e-participation rise as an alternative. This e-participation provides several advantages: connecting individuals with each other, governance processes, and their decision makers; stimulating participatory governance; and promoting transparency by making data available to the public [98]. However, while this participation system has advantages in developed countries, its applicability in developing countries,

particularly in Africa, would have organizational and technological limitations [99,100] such as access to the network internet. Similar findings of difficulties with e-participation by populations were noted in Cameroon [101].

Nevertheless, for a sustainable solution, local communities should have access to microcredit to engage in other income-generating activities and thus reduce pressure on forest resources [102]. However, the system of supervision of local communities by NGOs and provincial public services, which otherwise organize their activities according to intermittent funding from external donors and the Congolese government, would be weakened. This may jeopardize the sustainability and continuity of sustainable and participatory management actions [103].

## 5. Conclusions

This study assessed the perception of local communities on the management of *miombo* woodlands in the rural area of Lubumbashi, through surveys of 945 heads of households.

The results show that local communities perceive soil fertility loss and deforestation as major environmental problems due to their heavy dependence on forest resources. Indeed, these major environmental problems result from anthropogenic activities, mainly shifting agriculture and charcoal production, commonly practiced in the Lubumbashi rural area. In addition, regardless of the activity, adult males with a high level of education perceive environmental problems more accurately because of their repeated intrusion into the forest and awareness of current environmental issues. This result is certainly derived from the social division of labor and schooling by gender in Lubumbashi rural area especially and as is usually in the African rural areas. Finally, due to the lack of inclusion of local communities' perception in the forest management programs, these results reveal poor participation of local communities in the activities of provincial public services and NGOs, especially since these activities remain a low priority and do not correspond to the expectations of these communities.

Therefore, the actions to be carried out must result from concerted and inclusive planning, for their better appropriation and the sustainable management and resilience of *miombo* woodlands to anthropic pressures.

**Supplementary Materials:** The following supporting information can be downloaded at: https://www.mdpi.com/article/10.3390/f14040687/s1.

**Author Contributions:** Writing—original draft, D.-d.N.N.; writing—review, H.K.M.; B.K.W.N.K., K.R.S. and F.M.; supervision and writing-original draft, Y.U.S.; W.M.K. and J.B. All authors have read and agreed to the published version of the manuscript.

**Funding:** This research was funded by the Academy of Research and Higher Education (ARES-CCD, Belgium) via the Development Research Project: "Capacity Building for the Sustainable Management of the *miombo* woodlands through the Assessment of the Environmental Impact of Charcoal Production and the Improvement of Forest Resource Practices" (PRD CHARLU).

**Institutional Review Board Statement:** Not applicable.

**Informed Consent Statement:** Not applicable.

**Data Availability Statement:** The data related to the present study will be available upon request from the interested party.

**Acknowledgments:** The authors would like to thank the Academy of Research and Higher Education (ARES) and the PRD CHARLU for the financial support of this study through the doctoral fellowship granted to Dieu-donné N'tambwe Nghonda and Héritier Khoji Muteya. The local community members and authorities of the villages who took part in this study are warmly thanked for their availability.

**Conflicts of Interest:** The authors declare no conflict of interest.

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
