# Peer review of "Towards an Inclusive Approach to Forest Management: Highlight of the Perception and Participation of Local Communities in the Management of miombo Woodlands around Lubumbashi (Haut-Katanga, D.R. Congo)"

_forests, doi:10.3390/f14040687_

Round 1

Reviewer 1 Report

This paper examined the perception and participation of local communities in the management of miombo woodlands in Congo. The topic is interesting and timely. Here are my comments to improve the paper:

1- the title of the paper has put too much emphasis on the case study. Try to come up with a title that is more informative about what your paper is about, rather than the specific case study.

2- you need to revise your abstract. You need to include the major aspects of the entire paper in a prescribed sequence that includes: 1) the research problem(s) you investigated, or knowledge gaps that your paper addresses; 2) the overall purpose of the study; 3) the basic design of the study, i.e., methodology; 4) major findings or trends found as a result of your analysis; and, 5) a brief summary of your interpretations and conclusions. At the moment, some of these aspects are missing or under-developed.

3- In the introduction, make clearer what knowledge gaps you identified and how your research addresses them. Also, make the research objectives/questions clearer. Answer the “so what?” question. Why investigating such matter is important? End the introduction with an outline of the paper; what comes next?

4- The novelty/originality should be clearly justified that the manuscript contains sufficient contributions to the new body of knowledge from the international perspective.  What new things (new theories, new methods, or new policies) can the paper contribute to the existing international literature? This point must be reasonably justified by a Literature Review, clearly introduced in Introduction Section, and completely discussed in Discussion Section.

5- you need a new section on literature review. You need to acknowledge the existing literature on the issue and clearly identify the knowledge gap.

6- It would be helpful to include some discussions in your paper on participation and e-participation in the context of developing countries. In this discussion you could provide further discussions on the barriers to public participation and community engagement in decision making processes. Here are some recent references:

https://doi.org/10.1016/j.cities.2021.103281   

https://doi.org/10.4018/978-1-6684-3706-3.ch044

https://doi.org/10.1177/2399808317712515 

7- in the methodology, include type of questions you asked the interviewees. Also, include some of the questions to give the reader a sense of your questionnaire.

8- What are the limitations of your study?

9- how did you validate your data?

10- you need to refer back to the literature and previous studies in your result, discussion and conclusion sections.

11- how generalisable your findings are other places? Provide some discussions around the generalisability of your findings in the discussion section.

12- The conclusion could do more to tease out the wider resonance of the paper for the journal's international readership.

Author Response

In attachment, the response.

Best regards

Reviewer 2 Report

The manuscript entitled ‘Perception and participation of local communities in the management of miombo woodlands in the rural area of Lubumbashi 3 (Haut-Katanga, R.D. Congo)’ by Nghonda et al. attempts to test the hypothesis that local communities perceive deforestation as one of the main environmental problems, due to their high level of dependence on forest resources. The work is well executed and well written. However, I have some suggestion which will make the manuscript more scientifically sound and readable.

Abstract is well written; but there are some grammatical errors.

Keywords: Perception should be added as a keyword.

Introduction is well but there are too many references. Try to minimize the cited reference.

Study area: More information about the demography and status of living should be added.

Methodology is well written.

Questionnaire used in the study should be added as supplementary materials for replication of the study.

Results: This section is well written but should be framed as per the proposed hypothesis

 Line 215-216 Lwisha and 215 Nsela villages recorded many women; any specific reason.

Discussion: This section should start with the explanation of the proposed hypothesis

Author Response

In attachement, the response file.

Regards

Round 2

Reviewer 1 Report

Thank you for addressing my comments.